Exploring the phenotypic and molecular diversity of Calendula officinalis L. cultivars featuring varying flower types

Nucia Aleksandra 1
Ociepa Tomasz 1
Król Beata beata.krol@up.lublin.pl 2
Okoń Sylwia sylwia.okon82@gmail.com 1
1 Institute of Plant Genetics, Breeding, and Biotechnology, University of Life Sciences in Lublin , Lublin , Poland
2 Department of Industrial and Medicinal Plants, University of Life Sciences in Lublin , Lublin , Poland
Zhang Qianwen
Electronic publication date: 2025 Jan 6
Publication date: 2025
Volume: 13
Electronic Location ID: e18814
Received 2024 Oct 2; Accepted 2024 Dec 13
Copyright: ©2025 Nucia et al.
Copyright year: 2025
Copyright holder: Nucia et al.
License: This is an open access article distributed under the terms of the Creative Commons Attribution License, which permits unrestricted use, distribution, reproduction and adaptation in any medium and for any purpose provided that it is properly attributed. For attribution, the original author(s), title, publication source (PeerJ) and either DOI or URL of the article must be cited.
License URL: https://creativecommons.org/licenses/by/4.0/

Keywords: Genetic diversity, Morphology, Potmerigold, SCoT

Funding: The authors received no funding for this work.

==============================
Pot marigold is an ornamental plant of great importance in pharmacy and cosmetology. However, there is limited published information on the diversity of the species at both the morphological and genetic levels. This paper aimed to determine the morphological and genetic diversity of selected marigold varieties. The research focused on varieties with different flower colours, and the most important morphological features of flowers and plant height were analysed. Genetic diversity analyses were performed using start codon target (SCoT) markers. The correlation coefficients between the analysed morphological features of the studied marigold varieties showed a significant positive relationship between plant height, the number of inflorescences per plant and inflorescence diameter. Genetic analyses grouped the studied varieties according to the colour of their flowers. The results obtained in our work indicate that, despite the variability observed in morphological features, the gene pool of the analysed marigold varieties is limited. This information can be valuable for marigold breeders, particularly for selecting varieties suitable for crossbreeding in breeding programs. Additionally, it offers insights into the genetic resources of the species and highlights the importance of their bioconservation.

Introduction

Pot marigold (Calendula officinalis L.) is an annual herbaceous plant that is cultivated worldwide for pharmaceutical, cosmetic, and decorative purposes (Lim, 2014). The herbal raw material of marigolds includes ligular flowers (Calendulae floss) or whole flower heads (Calendulae anthodium), which are a rich source of many compounds with healing properties (including triterpene saponins, carotenoids, flavonoids, essential oil, tannins) and have a wide range of pharmacological activity (Khalid & Da Silva, 2012; Szopa et al., 2020; Shahane et al., 2023). Industrial interest in this plant increased after it was discovered that marigold seeds contain about 20% of oil, and 60% of this oil consists of calendic acid (an isomer of linolenic acid) (Dulf et al., 2013; Król & Paszko, 2017). The chemical and biological properties of marigold seed oil facilitate its application in the chemical industry for the production of polymers and high-quality paints (Biermann et al., 2010) and in the pharmaceutical (Verma et al., 2018) and food industries (Fontes et al., 2017).

Distinctive features of the pot marigold’s appearance include an inflorescence composed of two types of flowers: ligular flowers on the external side and tubular flowers on the internal side (Szopa et al., 2020) (Fig. 1). The market of ornamental plants offers numerous varieties of pot marigolds of various plant heights, flower colours (from orange, through yellow, cream, apricot to reddish) and structure of the inflorescence (e.g., single, full, semi-double flower baskets; ligular flowers arranged in a tiled, needle-shaped arrangement, pompom) (Dzida et al., 2016). Pot marigold breeding focuses mainly on improving the traits such as ornamental value, flower yield, and herbal material quality (Salomé-Abarca et al., 2024). The most desirable varieties for the pharmaceutical industry are forms with numerous inflorescences, an intense orange colour, and the predominance of ligular flowers the most active compounds. The quality of inflorescences of Calendula officinalis cultivars, as a source of medicinal preparations, is influenced by the proportion of ligulate and tubular flowers (Ossipov et al., 2024) More flavonoid compounds are found in ligulate flowers (Król, 2012; Nurzyńska-Wierdak et al., 2015). The color of marigold flowers is mainly related to the content of carotenoids. Varieties with orange flowers contain more of these pigments than varieties with yellow flowers (Pintea et al., 2003; Nurzyńska-Wierdak, 2014). Genotypes with different plant heights, flower colours , and compact habitus are particularly suitable for decoration (Baciu et al., 2010).

Figure 1 Structure of the inflorescence C. officinalis.

However, despite the importance of the medicinal properties of pot marigold, there is limited information about the plant’s genetic diversity on both the morphological and molecular levels. Assessment of genetic diversity is an essential step in plant breeding programs, as it may assist in selecting proper cultivars and lines with higher diversity and better performance. Combining phenotyping and molecular evaluations provides more valuable information than their single assessment (Kiełtyka-Dadasiewicz et al., 2017; Kołodziej et al., 2018; Khalil, Ibrahim & Youssef, 2020). It has been suggested that the measurement of genetic diversity by molecular markers for breeding purposes should be based on functionally characterised genes or targeted genes because these may reflect functional polymorphism (Andersen & Lübberstedt, 2003; Poczai et al., 2013). One of the marker systems meeting this criterion is the start codon target (SCoT) method developed by Collard & Mackill (2009). The SCoT marker is based on the short-conserved region in plant genes flanking the ATG translation start codon. The technique is simple, low cost and highly polymorphic, and also provides extensive genetic information. Its primers are universal in plants (Khodaee et al., 2021; Al-Khayri et al., 2022; Igwe et al., 2022; Yeken et al., 2022).

The research aimed to evaluate the morphological and genetic diversity of marigold cultivars, which differ in flower colour. The results may be helpful in breeding for selection, hybridisation, biodiversity assessment, and conservation of diverse marigold gene pools.

Materials and Methods

Plant materials

The biological material was represented by 20 cultivars of C. officinalis that differed in the structure and colour of flowers, originating from different European providers (Table 1, Fig. 2). These genotypes were grown during the cropping season of 2020–2021 at the Experimental Farm of the University of Life Sciences in Lublin, Poland (51°15′00″N 22°34′00″E). The experimental field was located in an area with silt loam texture. The Ap horizon had a pH (KCl) of 6.8 and an organic matter content of 1.7%. Seeds were sown in the third decade of April in the amount of 8 kg ha−1 at a row spacing of 40 cm. After emergence, thinning was conducted, leaving about 40 plants per m2. The experiment was designed as a single-factor experiment, using the randomized block method. Three blocks were separated, in which 20 plots with varieties were randomly selected. The plots had an area of 2 m2. Each plot contained 80 plants.

Table 1 Origin and flower color of the analysed C. officinalis genotypes.

Code	Cultivar (country of origin and source)	Flower color	
		Ligulate	Tubular	
PM 1	Orange Gem (Holland, Muller Seeds)	Orange	Orange	
PM 2	Deja Vu (UK, Thompson & Morgan)	Orange/cream	Brown	
PM 3	Indian Prince (UK, Thompson & Morgan)	Dark orange	Brown	
PM 4	Apricot Beauty (Poland, PlantiCo )	Yellow-apricot	Yellow	
PM 5	Pink Surprise (Holland, Muller Seeds)	Yellow-pink	Yellow-brown	
PM 6	Cream Beauty (UK, Johnson & Son Ltd.)	Cream-coloured	Orange	
PM 7	Apricot Twist (France, Promesse de fleurs)	Cream-orange	Orange	
PM 8	Bon Bon Orange (UK, Benary)	Orange	Orange	
PM 9	Lemon Gem (UK Suttons)	Canary-yellow	Yellow	
PM 10	Greenheart Orange (France, Promesse de fleurs)	Dark orange	Lime-green	
PM 11	Orange Fire (UK, Compass Horticulture Ltd.)	Dark orange	Orange	
PM 12	Geisha Girl (Holland, Muller Seeds)	Orange-reddish	Brown	
PM 13	Flame Dancer (France, Vilmorin Garden)	Orange-reddish	Brown	
PM 14	Helios (Poland, Selecta)	Orange	Orange	
PM 15	Jowisz (Poland, Selecta)	Yellow	Yellow	
PM 16	Neptun (Poland, Selecta)	Yellow	Brown	
PM 17	Orange King (Holland, Horti Tops)	Orange	Orange	
PM 18	Sunset Buff (France, Vilmorin Garden)	Apricot-reddish	Brown	
PM 19	Santana (Poland, Polan)	Yellow	Yellow	
PM 20	Promyk (Poland, Polan)	Orange	Orange	

Figure 2 The inflorescences of 20 C. officinalis genotypes.

*Structure of the inflorescence. Full, F; semi-double, SD.

Morphological analysis

During the raw material harvest, the morphological features of the studied plants were assessed, characterizing 10 plants from each plot, in terms of their height and number of flower heads. The diameters of inflorescences and the number of rows of ligulate flowers were assessed based on 30 randomly selected inflorescences for each variety. The results are reported as the arithmetic mean of the two years.

The numerical results were statistically analysed using analysis of variance with Tukey’s confidence semi-intervals at a significance level of 0.05. The Pearson correlation coefficient between the morphological traits of the calendula plants was determined. Calculations were carried out in Statistica 9.0 and Excel 2021. A cluster analysis was conducted using the unweighted pair-group method with arithmetic mean (UPGMA) distance method using PAST (Hammer, Harper & Ryan, 2001).

Molecular analysis

Twenty seeds per cultivar were randomly planted in the phytotron chamber, and then the young leaves were harvested from grown plants from each cultivar using the bulk method. For DNA extraction, approximately 150 mg of leaf tissue was used, and the modified CTAB method described by Iqbal et al. (2013) was employed.

The analysis of genetic similarity was based on SCoT marker systems (Collard & Mackill, 2009). Reaction mixtures contained 1 × PCR Buffer (10 mM Tris pH 8.8, 50 mM KCl, 0.08% Nonidet P40) (Thermo Fisher Scientific, Waltham, MA, USA), 160 µM of each dNTP, 800 pM oligonucleotide primer, 1.5 mM MgCl2, 60ng of template DNA and 0.5 U Taq DNA Polymerase (Thermo Scientific), in a final reaction mixture of 10 µl. Amplification was carried out in a Biometra T1 thermal cycler programmed for 3 min at 94 ° C of initial denaturation, 35 cycles: 94 ° C—1 min, 50 ° C—45 1 min and 72 ° C—2 min, with a final extension at 72 ° C for 5 min. To verify the purity of reagents a negative control was added in each run. For reproducibility verification, every primers were tested twice. Amplification products were separated by electrophoresis in accordance with the conditions described by Kiełtyka-Dadasiewicz et al. (2017). PCR-amplified SCoT products were scored as present (1) or absent (0). In order to assess the polymorphism generated by SCoT markers, the following coefficients were calculated: the level of polymorphism of the primer, the relative frequency of polymorphic products (Belaj et al., 2001), the resolving power of the primer (Prevost & Wilkinson, 1999) and the polymorphic information content (PIC) (Anderson et al., 1993).

Genetic pairwise similarities (SI-similarity index) between studied genotypes were evaluated according to Dice’s formula after Nei & Li (1979). A cluster analysis was conducted using the UPGMA distance method was performed using PAST software (Hammer, Harper & Ryan, 2001).

Results

The calendula cultivars differed in their main agro-metrical traits. Plant height varied widely depending on the variety (Table 2). The highest height (73.5 cm) was recorded in the ‘Apricot Beauty’ variety, and the lowest height was reached by ‘Apricot Twist’ plants (35.2 cm). Based on the information provided by seed companies and literature (Nurzyńska-Wierdak et al., 2015; Mordalski et al., 2020), the varieties were divided into three groups: tall (over 60 cm), medium-tall (45–60 cm) and short (below 45 cm). Based on the results obtained, ten varieties were classified as tall (‘Indian Prince’, ‘Apricot Beauty’, ‘Cream Beauty’, ‘Greenheart Orange’, ‘Orange Fire’, ‘Flame Dancer’, ‘Helios’, ‘Jowisz’, ‘Orange King’, ‘Santana’), five varieties were characterised by medium height (‘Orange Gem’, ‘Deja Vu’, ‘Pink Surprise’, ‘Lemon Gem’, ‘Sunset Buff’) and five varieties were considered short (‘Apricot Twist’, ‘Bon Bon Orange’, ‘Geisha Girl’, ‘Neptun’, ‘Promyk’) (Table 2). The number of inflorescences and their diameter are essential features that differentiate marigold varieties, affecting herbal raw material’s decorative value and yield. The most inflorescences were formed by plants of the ‘Orange King’ variety (45.2 pieces ⋅ per plant), whose heads also reached the largest diameter (69.1 mm). The smallest number of inflorescences was formed by the ‘Orange Gem’ variety (17.7 pieces ⋅ per plant—Table 2), while inflorescences with the smallest diameter (44.4 mm) were produced by plants of the ‘Pink Surprise’ variety.

Table 2 Average of main morphological traits of 20 Calendula officinalis genotype.

Cultivar	Plant height (cm)	No. of inflorescences per plant	Diameter of inflorescences (mm)	No. of rows flower ligulate	
Orange Gem	47.7 ± 2.2fg	17.7 ± 1.4i	55.1 ± 2.4cd	8.2 ± 0.9c	
Deja Vu	51.3 ± 2.9ef	29.6 ± 3.1fg	47.1 ± 1.5e	5.1 ± 0.7h	
Indian Prince	62.6 ± 3.6cd	39.3 ± 3.5bc	65.5 ± 2.8a	8.3 ± 0.8c	
Apricot Beauty	73.5 ± 3.8a	44.2 ± 3.4a	59.0 ± 2.1bc	9.0 ± 1.7b	
Pink Surprise	53.5 ± 2.3ef	24.8 ± 2.3gh	44.4 ± 1.9f	5.2 ± 0.7h	
Cream Beauty	66.0 ± 3.0cd	40.3 ± 3.6b	55.6 ± 2.3c	6.0 ± 1.1f	
Apricot Twist	35.2 ± 2.1i	28.6 ± 2.9f	47.8 ± 2.1e	5.5 ± 0.7h	
Bon Bon Orange	42.0 ± 2.6g	30.0 ± 3.0e	45.8 ± 1.8ef	3.0 ± 0.5i	
Lemon Gem	46.1 ± 2.7gh	24.3 ± 2.7h	52.1 ± 1.9de	5.1 ± 0.9h	
Greenheart Orange	72.4 ± 3.9ab	35.4 ± 2.8c	63.9 ± 2.7ab	7.2 ± 1.2de	
Orange Fire	64.7 ± 3.1cd	36.7 ± 2.6bc	55.7 ± 2.2cd	5.1 ± 0.8h	
Geisha Girl	44.7 ± 2.4gh	34.8 ± 3.1c	48.7 ± 1.5ef	9.1 ± 1.3b	
Flame Dancer	67.6 ± 3.8abc	39.0 ± 3.3b	51.1 ± 2.1d	7.3 ± 1.1d	
Helios	67.7 ± 3.5abc	39.2 ± 3.0b	66.0 ± 2.8a	9.2 ± 1.0ab	
Jowisz	61.3 ± 3.4cd	37.9 ± 2.6bcd	65.3 ± 3.1a	7.5 ± 0.9d	
Neptun	44.3 ± 1.8g	34.0 ± 2.9de	48.9 ± 1.8e	6.6 ± 0.7ef	
Orange King	61.6 ± 3.1cd	45.2 ± 3.1a	69.1 ± 3.5a	9.8 ± 1.3a	
Sunset Buff	58.2 ± 2.9de	30.1 ± 2.2ef	51.2 ± 2.2d	5.7 ± 0.6gh	
Santana	64.5 ± 3.2cd	35.4 ± 2.4cd	64.9 ± 3.3a	6.2 ± 0.7fg	
Promyk	40.2 ± 2.5hi	25.5 ± 1.9gh	48.4 ± 1.4e	7.8 ± 0.9cd	
Notes.

Data represents means of two years ± standard deviation (SD).

Mean values within a column by different letters are significantly different at P ≤ 0.05.

In marigolds, inflorescences are distinguished by the number of rows of ligular flowers, into single (1–2 rows of ligular flowers), semi-double (3–5) and full (six or more rows) (Mitu et al., 2020). In the current study, 13 cultivars produced full inflorescences (with 6–9 rows of petals) (‘Orange Gem’, ‘Indian Prince’, ‘Apricot Beauty’, ‘Cream Beauty’, ‘Greenheart Orange’, ‘Geisha Girl’, ‘Flame Dancer’, ‘Helios’, ‘Jowisz’, ‘Neptun’, ‘Orange King’, ‘Santana’, ‘Promyk’) and seven produced semi-double inflorescences (with 3–5 rows of petals) (‘Deja Vu’, ‘Pink Surprise’, ‘Apricot Twist’, ‘Bon Bon Orange’, ‘Lemon Gem’, ‘Orange Fire’, ‘Sunset Buff’) (Table 2, Fig. 2).

Based on the results of the morphological studies, a cluster analysis was performed using the UPGMA method. The studied marigold genotypes formed two groups on the dendrogram (Fig. 3). The first group includes ten varieties characterised by high growth and a large number of inflorescences (over 35 per plants). Considering the diameter of the inflorescences, two subgroups were distinguished within the first group, where, the first subgroup includes six varieties with heads over 60 mm (‘Greenheart Orange’, ‘Indian Prince’, ‘Jowisz’, ‘Santana’, ‘Helios’, ‘Orange King’) and the second subgroup contains four varieties (‘Apricot Beauty’, ‘Cream Beauty’, ‘Orange Fire’, ‘Flame Dancer’) with slightly smaller heads (diameter 50–60 mm). The second group of clusters also included ten varieties that formed two subgroups. The first cluster included low varieties with small flower heads (diameter below 50 mm) (‘Apricot Twist’, ‘Promyk’, ‘Bon Bon Orange’, ‘Geisha Girl’, ‘Neptun’) and the second cluster included medium-tall varieties with medium-sized (50–60 mm) flower heads (‘Lemon Gem’, ‘Orange Gem’, ‘Sunset Buff’) or small (below 50 mm) flower heads (‘Deja Vu’, ‘Pink Surprise’).

Figure 3 UPGMA dendrogram of twenty C. officinalis cultivars based on morphological traits.

PM1-PM20 marigold variety codes. The assignment of codes to the appropriate varieties is given in Table 1.

The correlation coefficients between the analysed morphological features of the studied marigold varieties showed a significant positive relationship between plant height, the number of inflorescences per plant (r = 0.693), and inflorescence diameter (r = 0.679) (Table 3).

The DNA extraction and amplification methodology proved to be efficient in detecting polymorphism for the C. officinalis cultivars. Of the 30 SCoT primers tested, ten were selected and used in the study of the genetic diversity among C. officinalis genotypes. PCRs using selected primers amplified 100 clear bands. Among these, 61 (61.00%) were polymorphic. Number of amplified fragments varied from four to 13, averaging 5 per primer and 10 per genotype. The number of polymorphic bands ranged from 2 (SCoT 26, SCoT 27, SCoT 31) to 13 (SCoT 12), with a mean of 3.05 per primer and 6.1 per genotype (Table 4). The highest primer diversity showed SCoT 12 (86.67%). The lowest level of primer diversity showed SCoT31 (18.18%). The PIC values varied from 0.06 (SCoT31) to 0.30 (SCoT12), averaging 0.19. The Rp value of the ten primers varied between 1.7 (SCoT 26) to 13.2 (SCoT21).

Table 3 Correlation coefficients between measured traits of C. officinalis.

Morphological traits	1	2	3	4	
1	Plant height	1				
2	No. of inflorescences	0.693*	1			
3	Diameter of inflorescences	0.679*	0.512	1		
4	Rows of flower ligulate	0.349	0.423	0.485	1	
Notes.

* Significant at the 0.05 probability level.

Table 4 Characteristics of selected SCoT primers.

Lp.	Primer no.	Sequence 5′–3′	Number of products	Frequency of polymorphic products	Resolving power of the primer	PIC	
			Total	Polymorphic	Monomorphic				
1	SCoT 12	ACGACATGGCGACCAACG	15	13	2	86.67	12.1	0.30	
2	SCoT 21	ACGACATGGCGACCCACA	13	11	2	84.62	13.2	0.26	
3	SCoT 24	CACCATGGCTACCACCAT	4	3	1	75.00	3.4	0.15	
4	SCoT 25	ACCATGGCTACCACCGGG	7	5	2	71.43	1.9	0.19	
5	SCoT 26	ACCATGGCTACCACCGTC	11	2	9	18.18	1.7	0.06	
6	SCoT 27	ACCATGGCTACCACCGTG	5	2	3	40.00	2.5	0.15	
7	SCoT 31	CCATGGCTACCACCGCCT	11	2	9	18.18	2.4	0.06	
8	SCoT 71	CCATGGCTACCACCGCCG	12	8	4	66.67	6.5	0.21	
9	SCoT 84	ACGACATGGCGACCACGT	10	8	2	80.00	7.7	0.27	
10	SCoT 90	CCATGGCTACCACCGGCA	12	7	5	58.33	4.3	0.21	
Total	100	61	39	61.00			
Average/primer	5	3.05	1.95				
Average/genotype	10	6.1	3.9				

The genetic similarity matrices were produced based on SCoT using Dice’s coefficient. Genetic similarity indexes were estimated between 0.613 for PM5 and PM20 cultivars and 0.959 for PM14 and PM20. The mean genetic similarity was calculated at 0.754. PM5 cultivar (0.726) showed the lowest genetic similarity to the rest of the analysed genotypes , and cultivar PM7 (0.77) was the most similar to the other genotypes (Table 5).

Table 5 Genetic similarity indices calculated based on ScoT marker polymorphisms for the analyzed marigold cultivars.

Cultivar	Orange Gem	Deja Vu	Indian Prince	Apricot Beauty	Pink Surprise	Cream Beauty	Apricot Twist	Bon Bon Orange	Lemon Gem	Greenheart Orange	Orange Fire	Geisha Girl	Flame Dancer	Helios	Jowisz	Neptun	Orange King	Sunset Buff	Santana	Promyk	
Orange Gem																					
Deja Vu	0.813																				
Indian Prince	0.866	0.800																			
Apricot Beauty	0.684	0.678	0.719																		
Pink Surprise	0.619	0.631	0.691	0.920																	
Cream Beauty	0.862	0.760	0.817	0.691	0.642																
Apricot Twist	0.884	0.772	0.841	0.741	0.679	0.885															
Bon Bon Orange	0.839	0.754	0.843	0.685	0.673	0.786	0.862														
Lemon Gem	0.689	0.667	0.690	0.906	0.863	0.696	0.763	0.708													
Greenheart Orange	0.862	0.781	0.882	0.718	0.673	0.829	0.884	0.871	0.739												
Orange Fire	0.785	0.875	0.740	0.650	0.637	0.764	0.775	0.790	0.655	0.785											
Geisha Girl	0.688	0.667	0.705	0.786	0.759	0.644	0.726	0.706	0.737	0.704	0.736										
Flame Dancer	0.672	0.700	0.706	0.844	0.800	0.661	0.744	0.672	0.775	0.705	0.689	0.906									
Helios	0.784	0.862	0.754	0.643	0.630	0.763	0.742	0.807	0.614	0.800	0.880	0.733	0.684								
Jowisz	0.661	0.672	0.678	0.870	0.846	0.702	0.750	0.696	0.873	0.694	0.661	0.793	0.796	0.655							
Neptun	0.705	0.717	0.739	0.881	0.876	0.696	0.727	0.690	0.865	0.721	0.672	0.786	0.789	0.684	0.903						
Orange King	0.870	0.791	0.859	0.678	0.632	0.839	0.831	0.848	0.667	0.870	0.763	0.730	0.732	0.825	0.672	0.715					
Sunset Buff	0.694	0.672	0.712	0.796	0.788	0.667	0.717	0.696	0.764	0.727	0.694	0.914	0.885	0.724	0.786	0.814	0.738				
Santana	0.723	0.688	0.693	0.855	0.832	0.699	0.729	0.710	0.824	0.738	0.723	0.800	0.770	0.704	0.826	0.852	0.702	0.810			
Promyk	0.797	0.857	0.736	0.626	0.613	0.777	0.740	0.820	0.615	0.781	0.891	0.715	0.667	0.959	0.655	0.667	0.837	0.706	0.719		
Mean similarity	0.763	0.745	0.762	0.756	0.726	0.746	0.778	0.761	0.743	0.777	0.746	0.749	0.747	0.750	0.747	0.763	0.768	0.753	0.758	0.746	

A genetic similarity matrix was applied for cluster analysis using the UPGMA method (Fig. 4). The 20 C. officinalis cultivars were grouped into three major clusters based on bootstrapping. Group A contained three red–flower cultivars: ‘Geisha Girl’, ‘Flame Dancer’ and ‘Sunset Buff’. In group B, cultivars with yellow-coloured flowers: ‘Apricot Beauty’, ‘Pink Surprise’, ‘Lemon Gem’, ‘Jowisz’, ‘Neptun’, and ‘Santana’ were grouped. Group C included ‘Orange Fire’, ‘Helios’, ‘Promyk’, ‘Deja Vu’, ‘Cream Beauty’, ‘Apricot Twist’, ‘Orange Gem’, ‘Bon Bon Orange’, ‘Indian Prince’, ‘Greenheart Orange’ and ‘Orange King’ with orange flowers.

Figure 4 The UPGMA analysis of the 20 C. officinalis cultivars groups reflecting flower colors have been marked with appropriate colors.

PM1-PM20 marigold variety codes. The assignment of codes to the appropriate varieties is given in Table 1.

Discussion

The C. officinalis species is characterised by high morphological variability (Król, 2012; Baciu et al., 2013). According to Nurzyńska-Wierdak et al. (2015), plant height is a typical cultivar feature, but it is also influenced by environmental factors. The differences in the height of marigolds depending on the genotype are reported by Baciu et al. (2013) (height ranged from 22 cm to 71.2 cm), Massoud et al. (2020) (range 52.2–70.5 cm) and Król & Paszko (2017) (range 46–75 cm). In this study, the height of the marigolds ranged from 35.2 cm to 73.5 cm, which confirms the opinions of Nurzyńska-Wierdak et al. (2015) about the existence of different plant size groups among marigold varieties. However, this feature can be differentiated to some extent by, among others, habitat conditions (Ratajczak et al., 2016), mineral fertilisation (Duda et al., 2010; Arab et al., 2015) and plant density (Mirzaei et al., 2016). The number and diameter of marigold inflorescences are plant features significant for the cultivation for ornamental purposes or as a herbal raw material. These features are mainly determined by genotype (Król, 2012; Baciu et al., 2013; Nurzyńska-Wierdak et al., 2015) but are also impacted by the influence of natural and agrotechnical factors. In the literature data, these marigold feature values range extensively, i.e., the number of heads from 15 (Naguib, Khalil & Sherbeny, 2005) to 140 pieces ⋅ plant-1 (Khalid & Zaghloul, 2006) and the inflorescence diameter from 0.86 cm (Ganjali et al., 2010) to 10.6 cm (Nurzyńska-Wierdak et al., 2015). Significant discrepancies in the discussed values result from climatic conditions and experimental factors. The values obtained in the present study regarding the number of heads ranged from 17.7 to 45.2 pieces ⋅ plant-1 and the inflorescence diameter from 44.4 mm to 69.1 mm, confirming the high genetic and ecotypic variability of marigold. The type of inflorescence and colour of flowers are essential for medicinal purposes, and genotypes with large, abundant flowers and intense orange colour are the most commonly used (Baciu et al., 2010; Mitu et al., 2020). Among the tested varieties, the most suitable for pharmaceutical cultivation can be identified based on these features.

Choosing suitable molecular markers for genetic diversity analysis is crucial in using available genetic resources to expand a plant’s gene pool. Several types of molecular markers differ in their basis and target region of the genome. In most studies, markers employed were derived from random genomic regions and were phenotypically neutral. The fast-paced genomic research has facilitated researchers to move towards markers with target coding regions of the genome instead of random regions (Kage et al., 2015). In the current study, start codon target (SCoT) markers were used to evaluate the genetic diversity of 20 C. officinalis cultivar varieties that differed in the structure and colour of flowers. This method generates plant DNA markers based on the short and conserved regions flanking the ATG start codon in plant genes. Many plant species have been used in this fast and simple method to explore the genetic diversity and relationships among different plant genotypes (Rai, 2023). In many cases, SCoT exposes a higher percentage of polymorphism (%P) than other markers (Singh et al., 2017; Khalil, Ibrahim & Youssef, 2020). SCoT markers generate a high level of polymorphism (%P) and resolving power of the primer (Rp), which were shown from many plant species, for example, %P = 60% for Indian poppy genotypes (Srivastava et al., 2020), %P = 77.21% and Rp = 6.108 for whip grass (Guo et al., 2014), 76.19% for mango cultivars (Luo et al., 2010) and %P = 100% and Rp = 10.96 for durum wheat genotypes (Etminan et al., 2016). Khalil, Ibrahim & Youssef (2020) obtained a slightly lower level of polymorphism (48.57%) and resolving power(1.71) by analysing the genetic diversity of doum palm landraces. In the current study, selected SCoT primers generated 100 bands, of which 61 (61.00%) were polymorphic among analysed pot marigold cultivars. The average Rp value for tested cultivars was 5.57. A great polymorphism detection power in different plant species (closer or further related) has proven that SCoT markers are helpful in genetic diversity studies.

There are few reports in the available literature on analysing marigold genetic diversity. The available publications are mainly based on the use of RAPD markers, which are increasingly rarely used due to numerous disadvantages. However, in many cases, these markers provide preliminary information on genetic diversity. Baciu et al. (2013) used RAPD markers to estimate the genetic diversity of C. officinalis genotypes in 13 countries. Baciu et al. (2010) analysed the genetic diversity of 45 genotypes of the Calendula genera. Giancarla et al. (2018), based on RAPD, estimated genetic diversity among 19 C. officinalis cultivars from Romania. Xu, Li & Ge (2001) used the RAPD markers to discriminate yellow-flowered and orange-flowered calendula genotypes. All of these authors used randomly amplified markers, which are not correlated with target genes. However, the analysis of the results showed that some phenotype traits are associated with genetic diversity. For example, Baciu et al. (2013) found that some genotypes with the same types of flowers can be grouped based on RAPD results. Comparing the results of the RAPD-based dendrogram with the peculiarities of plants Baciu et al. (2010) also found some similarities at the molecular and phenotype level observation. In their experiment, some genotypes in the same dendrogram subgroup have similar phenotype traits. Zagorcheva et al. (2022) used sequence-related amplified polymorphism (SRAP) markers to estimate genetic diversity among locally growing C. officinalis wilding plants in Bulgaria. They proved that these markers provide valuable data to estimate genetic diversity and identify high polymorphism among tested plants. The current study used SCoT markers to assess the genetic diversity of commercially grown marigold varieties. As a result, 20 analysed cultivars were grouped in three clusters regarding the flower colour. This suggests that SCoT markers are more valuable for genetic diversity analysis of genotypes with similar phenotype traits. Moreover, the authors of the present study used markers to divide cultivars into subclusters according to their origin, reflecting distinct genetic pools of varieties from different breeding companies. Twenty analysed calendula cultivars showed high genetic similarities (average 0.848), which indicates a narrow gene pool of the analysed species. A high level of similarity was also noted by Baciu et al. (2013).

Conclusions

Morphological and genetic characteristics of plant varieties provide valuable information for breeders and allow for better management of a given species’ available resources. The presented work assessed the morphological and genetic diversity of selected marigold varieties with different flower colours.

The studies showed significant morphological variability, especially regarding plant height, number of inflorescences and flower diameter. The varieties were divided into three groups: tall, medium, tall and short, emphasising the diversity of traits depending on the genotype. Plant height and the number and size of inflorescences are mainly determined by genotype and environmental factors such as habitat conditions and cultivation practices.

SCoT markers were used to assess genetic diversity, and the analysed varieties showed a significant level of polymorphism. This method effectively detected genetic variability, which is crucial for expanding the gene pool and improving cultivation practices.

However, despite the observed variability, the average genetic similarity coefficient was relatively high, which indicates a narrow gene pool. Further efforts to introduce genetic diversity through breeding programs may be necessary to increase the resistance and adaptability of C. officinalis varieties. The study’s results provide valuable information for breeders and producers, emphasising the importance of considering both morphological traits and genetic diversity when selecting varieties for ornamental cultivation and pharmaceutical use.

Supplemental Information

Data S1 Raw data

Additional Information and Declarations

Competing Interests

Author Contributions

Data Availability

The authors declare there are no competing interests.

Aleksandra Nucia conceived and designed the experiments, performed the experiments, authored or reviewed drafts of the article, and approved the final draft.

Tomasz Ociepa performed the experiments, prepared figures and/or tables, and approved the final draft.

Beata Król conceived and designed the experiments, performed the experiments, analyzed the data, prepared figures and/or tables, and approved the final draft.

Sylwia Okoń conceived and designed the experiments, performed the experiments, analyzed the data, prepared figures and/or tables, authored or reviewed drafts of the article, and approved the final draft.

The following information was supplied regarding data availability:

The raw data is available in the Supplemental File.

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
