# Peer review of "Exploring the phenotypic and molecular diversity of Calendula officinalis L. cultivars featuring varying flower types"

_PeerJ, doi:10.7717/peerj.18814_

## Round 0.1 · original submission · Major Revisions

Thanks for submitting your manuscript to PeerJ. Please revise the manuscript carefully according to the comments made by reviewers. Thanks.

Reviewer 1 ·

Basic reporting

no comment

Experimental design

no comment

Validity of the findings

no comment

Additional comments

The research assessed the morphological and genetic diversity of marigold cultivars. This study demonstrates clear and effective use of English throughout and provides sufficient background. However, in my opinion, the experimental design is not rigorous, and the data provided is not comprehensive and sufficient to meet the research purpose. Here are some details,
1. What was the experimental design? How many blocks were there?
2. Why do molecular experiments need to be replanted? Leaf samples can be collected from the previous experiment.
3. Add statistical results for results.
4. Some results are incorrect.

Annotated reviews are not available for download in order to protect the identity of reviewers who chose to remain anonymous.

Reviewer 2 ·

Basic reporting

This work is based on pot marigold which has important pharmaceutical, cosmetology, and ornamental values. The proposed research aim was to use SCoT markers in dissecting plant genetic diversity in this population to inform future breeding, and the main traits of interest were on morphology and flower color. This work is filling the gap of a paucity of research on identifying genetic diversity of this specialty crop.

Experimental design

Line 83- 101 Twenty cultivars of pot marigold were used in this trial. Is your experimental design in the field trial the same as in the molecular analysis? There is a lack of description in number of reps, blocks, and randomization of the field experiment.

Statistical analysis
The results from ANOVA are not indicated in the manuscript
Table 2 The clarity of data will be improved with post-hoc pairwise comparisons, with letter groups assignment.

Validity of the findings

Line 65 description of the comparison between the markers system is vague. Will you please add more details of why the SCoT method was more appropriate than the rest?

Table 1 There are two main categories of flower types, ligulate and tubular. It would be nice to have illustrations to show the readers what the differences are between the two types of flowers.

Figure 1 Can you name each of the inflorescence structure per genotype in the figure?

Line 216-217 How are the variations in plant morphology be directly linked to pharmaceutical values? There is no wet lab analysis in this experiment to support this argument.

Additional comments

Line 2 remove period at the end of the main title

Line 137-138 Double check the language (not English) to ensure clear information

Reviewer 3 ·

Basic reporting

no comments

Experimental design

Please see comments below

Validity of the findings

Please see the below comments and the conclusion is reluctant.

Additional comments

This manuscript describes using SCoT markers to interpret the population structure of 20 marigold varieties. The authors first describe the plant height, the number of inflorescence per plant, the diameter of inflorescence, and the number of row of flower ligulates. After that, the authors use a clustering method to sort out the structure of phenotypic traits among 20 cultivars. Overall, the manuscript is very descriptive and I think it needs large improvement. Please see the points below.


Major concerns:
1. What’s the real biological nature to cluster the phenotypic traits, any biological connections with the clustering results from genotyping results (Figure 3)?
2. How do authors explain the discrepancy between the clustering results (Figure 2) and genotyping cluster results (Figure 3)?
3. In figure 3, are these three major groups correlating with the origin of cultivars like geological location, pedigree information?
4. more data should support your genotyping results and the clustering results of phenotypic traits do not contribute significant biological meaning unless the authors can provide more data or insight to support them.



Minor points:
Line 137-138 should be English version instead of native language
Line 177 miss period symbol after “(86.67%)”
Line 178 0,06 (SCoT31) should 0.06 (SCoT31)

---

## Round 0.2 · Minor Revisions

Please provide a suitable response/revisions to each question raised by the reviewer.

Reviewer 1 ·

Basic reporting

no comment

Experimental design

no comment

Validity of the findings

no comment

Additional comments

1. The experimental design is still confusing. Please specify how many blocks there are and how many plants of each cultivar are in each block.
2. In the display of results, select one of the cultivar name or code (PM) and keep it consistent, and add the cultivar name of the code in the table and figure legends.

Reviewer 2 ·

Basic reporting

No comment

Experimental design

No comment

Validity of the findings

No comment

Additional comments

No comment

Reviewer 3 ·

Basic reporting

no comment

Experimental design

no comment

Validity of the findings

no comment

Additional comments

I think the authors address most of my concerns and i have no additional comments.

---

## Round 0.3 · Minor Revisions

Regarding the codes used in dendrograms, please at least mention in the legends of Figures 3&4 that the interpretation of codes can be found in Table 1. Otherwise, it will be difficult for audience to understand the dendrograms.

Besides, please address the comments from the Section Editor: "I recommend adding a 'Conclusions' section.

In addition, the authors should proofread for style, e.g., eliminating phrases such as 'the results obtained are highly significant' and 'they are also of great importance.'"

Thanks.

---

## Round 0.4 · accepted · Accept

The abstract was updated in the revised manuscript but was not updated in the submission system (see PDF page 1). Please address this issue during the proofread stage.